# Review of High-Frequency Ultrasounds Emulsification Methods and Oil/Water Interfacial Organization in Absence of any Kind of Stabilizer

**DOI:** 10.3390/foods11152194

**Published:** 2022-07-23

**Authors:** Louise Perrin, Sylvie Desobry-Banon, Guillaume Gillet, Stephane Desobry

**Affiliations:** 1Laboratoire d’Ingénierie des Biomolécules (LIBio), Université de Lorraine, 2 Avenue de la Forêt de Haye, CEDEX, BP 20163, 54505 Vandœuvre-lès-Nancy, France; sylvie.desobry@univ-lorraine.fr (S.D.-B.); stephane.desobry@univ-lorraine.fr (S.D.); 2SAS GENIALIS, Route d’Achères, 18250 Henrichemont, France; g.gillet@genialis.fr

**Keywords:** emulsifier-free emulsion, emulsification processes, 1–5 MHz ultrasounds, oil/water interface, hydroxide ions adsorption

## Abstract

Emulsions are multiphasic systems composed of at least two immiscible phases. Emulsion formulation can be made by numerous processes such as low-frequency ultrasounds, high-pressure homogenization, microfluidization, as well as membrane emulsification. These processes often need emulsifiers’ presence to help formulate emulsions and to stabilize them over time. However, certain emulsifiers, especially chemical stabilizers, are less and less desired in products because of their negative environment and health impacts. Thus, to avoid them, promising processes using high-frequency ultrasounds were developed to formulate and stabilize emulsifier-free emulsions. High-frequency ultrasounds are ultrasounds having frequency greater than 100 kHz. Until now, emulsifier-free emulsions’ stability is not fully understood. Some authors suppose that stability is obtained through hydroxide ions’ organization at the hydrophobic/water interfaces, which have been mainly demonstrated by macroscopic studies. Whereas other authors, using microscopic studies, or simulation studies, suppose that the hydrophobic/water interfaces would be rather stabilized thanks to hydronium ions. These theories are discussed in this review.

## 1. Introduction

Emulsions are commonly used in many fields, such as food, cosmetic, pharmaceutical, agrochemical, paint, printing, as well as petroleum industries, and are thus used for drug delivery, nanomaterial preparation, or chemical reaction media [1,2,3,4].

Emulsions are metastable colloidal systems which tend to separate in different phases over time. Four main physicochemical mechanisms contribute, in combination or individually, to phase separation (Figure 1).

Gravitational separations, creaming, and sedimentation are reversible phenomena and correspond to the rise or fall of droplets, respectively, depending on dispersed phase density. Creaming and sedimentation velocities ν are determined by Stokes formula (Equation (1)):(1)$$\nu =\frac{{2\;g\;r}^{2}{(\rho }_{\mathrm{cont}}{\;-\;\rho }_{\mathrm{drop}})}{{9\;\eta }_{\mathrm{cont}}},$$
where r, g, ρ_cont_, ρ_drop_, and η_cont_ correspond to droplet radius, gravity acceleration, continuous phase density, droplet density, and continuous phase viscosity, respectively. Thus, to reduce these gravitational destabilizations, it is possible to reduce droplet size, reduce difference density between two phases, or to increase the continuous phase viscosity [5]. Flocculation is another reversible destabilization phenomenon, where droplets are gathering to form flocs or aggregates, due to attractive interactions between them. Droplets aggregation depends on surface electric charge, which is fluently determined by its ζ-potential [5,6]. Electric charge on droplet surface is structured in an electrical double layer. The first layer, called “Stern layer”, is constituted of a single ion layer which is immobile at the interface. This layer is then surround by a diffuse layer, “Gouy layer”, constituted of mobile ions [7]. To limit flocculation and to increase emulsions stabilization in the long-term, the ζ-potential magnitude should be greater than 20 mV. This can be achieved by using charged emulsifiers (as proteins or ionic surfactants) that promote electrostatic repulsions between droplets. Steric repulsions, favored by uncharged macromolecules’ (such as polysaccharides or non-ionic surfactants) presence at the interface, can also help to stabilize droplets and avoid flocculation [8]. Flocculation can then lead to coalescence, an irreversible phenomenon. Coalescence corresponds to the merge of droplets leading to bigger droplets formation until phase separation. The fourth destabilization mechanism is Ostwald ripening, which consists of droplets growth like coalescence. However, in this case, growth is due to the migration of dispersed phase molecules between droplets. This transfer is caused by the slight solubility of the dispersed phase in the continuous phase, which migrates between droplets. This phenomenon is governed by chemical potential difference between droplets of different size. Indeed, due to Laplace pressure, chemical potential of the dispersed phase is more important in smaller droplets than in larger ones, causing a mass transfer from smaller droplets to larger ones [5,6,9].

To facilitate emulsions formulation, droplet size reduction, and slow their destabilization, emulsifiers are currently used [4,10]. Emulsifiers are amphiphilic and contain both hydrophilic groups and hydrophobic tails, such as polysorbates and sorbitan derivatives [11] that are widely used in industries from petroleum to the cosmetic fields. Because of their chemical origin, most of the stabilizers are not biodegradable and can persist in the environment causing adverse effects on organisms [12,13,14,15]. Limiting stabilizers’ use or replacing chemical stabilizers by green and natural biomolecules, such as glycolipids, lipopeptides, as well as phospholipids, is challenging because it is difficult to obtain them at the industrial scale [12]. Pickering emulsions formulation with natural particles [16,17,18], as well as emulsifier-free emulsion formulation [2], i.e., emulsions formulated in absence of any stabilizers, such as proteins, polysaccharides, lipid-derived molecules, particles [6,8] represent other alternatives.

Emulsification requires energy input. The processes usually used are high-speed homogenizers, low-frequency ultrasounds generators, high-pressure homogenizers, or microfluidizers [8]. The first part of this paper will focus on the methods enabling the formulation of oil/water emulsions. In combination with current emulsification processes, a new high-frequency ultrasound method will be described as an additional step to increase the emulsions’ stability, without using a stabilizer [2,19,20]. However, the effect of high-frequency ultrasounds on emulsions’ stability is for the moment not fully explained.

A first line of study consists in understanding how the oil/water interface is organized in absence of any kind of stabilizer. Some authors assume that stability would be due to hydroxide ions’ (OH^−^) organization at the interfaces [21,22,23,24,25,26], whereas others suppose that the hydrophobic/water interfaces would rather be stabilized by hydronium ions (H_3_O^+^) [27,28,29,30]. Thus, the second objective of this paper will be to clarify these opposite views to understand how emulsifier-free emulsions are stabilized and then try to understand the effect of high-frequency ultrasounds on emulsions’ stability.

## 2. Emulsification and Stabilization Processes

### 2.1. Main Emulsification Processes

In general, making an emulsion requires four elements: (i) an aqueous phase, (ii) an oily phase, (iii) energy to deform droplets and break them down into smaller droplets, and (iv) an emulsifier to stabilize the oil/water interface formed. The emulsifier also has another role, which is to decrease interfacial tension. Physically, to break down droplets, energy provided must be more important than Laplace pressure (P_L_), which corresponds to the pressure difference between the concave and convex side of the curved interface. Laplace pressure is given by Equation (2):(2)$$P_{L}=\gamma \;(\frac{1}{R_{1}}+\frac{1}{R_{2}}),$$
with γ the interfacial tension, and R_1_ and R_2_ the main curvature radii. Thus, by reducing interfacial tension, Laplace pressure decreases and therefore energy required for emulsification is less important [31].

Thus, emulsification can require high-energy or low-energy processes depending on emulsion composition, and emulsifier nature and content. As usual emulsification methods have already been studied and reviewed many times [6,8,32,33,34,35,36,37], we will briefly present their main characteristics in this part.

#### 2.1.1. High-Energy Emulsification Methods

High-energy emulsification methods are widely used in industry despite their cost. The dissipated energy results in an efficient reduction of droplet size in emulsion but it can also affect the thermolabile active ingredients and thus, required temperature-controlled device [38].

##### High-Speed Homogenization

High-speed homogenization is a mechanical stirring method based on the rotor/stator system (Figure 2A). Droplets’ breakage is due to inertia and shear force in turbulent flow. The disc system (Figure 2B) is similar to the rotor/stator system and adapted to high viscosity fluids and/or formulation with sensitive components.

These methods are generally used to produce macroemulsion, pre-emulsion, or premix, before using other emulsification methods [34,39,40].

##### Low-Frequency Ultrasounds

Ultrasounds are produced by piezoelectric transducers that convert electrical pulses into acoustic energy waves (Figure 3). Low-frequency ultrasounds, between 20 and 100 kHz, correspond to high energy from 10 to 1000 W/cm^2^ [41]. Acoustic waves induce cavitation phenomenon which produce gas bubbles. These bubbles grow under acoustic wave effect, oscillating between compression and rarefaction phases, until reaching a critical size where they collapse violently, causing high temperature and pressure generation in their local environment (Figure 4). Thus, cavitation effect generates forces which break oil droplets into smaller ones [42].

A recent publication aimed to describe the ultrasonic emulsification phenomenon [43]. When cavitation bubbles generated by acoustic waves collapse, a micro-jet is produced, pushing water molecules into an oily phase and forming water-in-oil (W/O) emulsion. Then, due to Rayleigh–Taylor instability, droplets containing W/O emulsion are pulled apart from the oily phase into water phase. Under ultrasounds’ effect, these droplets are broken into smaller droplets until oil-in-water (O/W) emulsion is formed (Figure 5) [43].

Low-ultrasonic emulsification has several advantages such as giving small droplets, low size polydispersity, and high ζ-potential value, resulting in the structural stability of emulsion with time. However, the bubble collapse that is specific of this process produces a lot of heat that can induce free radicals’ formation from water molecules and then deleterious oxidation of emulsion components (Equation (3) [42].
H_2_O → H^•^ + OH^•^(3)

##### High-Pressure Homogenization

High-pressure homogenization (HPH) consists of applying high pressure to force the passage of oily and aqueous phases in a premix, through an orifice between a valve seat and a forcer (Figure 6).

During HPH treatment, two opposite mechanisms occur impacting oily droplet size. First, droplets’ fragmentation is caused by the combination of (i) shear forces in the orifice, or in orifice boundaries, (ii) cavitation forces, due to pressure difference when passing from the inside to outside orifice, taking place in the orifice, and (iii) turbulent flow due to fluid passage through the orifice. Secondly, collisions between droplets cause coalescence and droplet enlargement. To promote fragmentation, it is necessary to use emulsifiers with a high adsorption rate, or to increase number of passages through the valve.

This method gives droplets with a size ranging from 50 nm to 5 µm, depending on emulsion composition and process conditions, such as temperature, pressure (from 50 to 500 MPa), cycles number, as well as narrow orifice geometry.

Emulsification by HPH is a rapid process that can be carried out in continuous flow. Cold HPH has the advantage to treat thermosensitive compounds. However, this expensive process can only be used with medium and low viscosity emulsions and does not allow to treat shear-sensitive compounds [34,44].

##### Microfluidization

Similar to HPH, microfluidization uses high pressures to force emulsion, or premix, to pass through a narrow orifice, but in this case, emulsion is divided into two streams. The two emulsion jets formed coming from two channels collide with each other in an interaction chamber (Figure 7).

The droplet size decrease is essentially due to shear, inertia, and cavitation forces, which enable to obtain very fine droplets. Droplet size depends on pressure, cycle number, and emulsifier concentration, and can be easily controlled.

Microfluidizer could produce narrower and smaller droplet size than HPH. However, this process is difficult to use for large-scale production, and as HPH, is expensive [32,34,40,45].

#### 2.1.2. Low-Energy Emulsification Methods

##### Membrane and Microchannel Emulsifications

Membrane emulsification consists in forcing dispersed phase passage under pressure through a membrane containing uniform pores, towards the continuous phase flowing tangentially to the membrane (Figure 8).

Membrane nature depends on emulsion type. O/W emulsions are emulsified with hydrophilic membrane, whereas for W/O emulsions, membrane is hydrophobic, so that the dispersed phase does not wet the membrane.

Pressure applied depends on membrane pore size, interfacial tension, and production objectives. High pressures favor large droplets’ formation while low pressures decrease production rate.

Even though this process needs energy input to form droplets, membrane emulsification requires much less energy than other processes mentioned above, and size droplets’ decrease is not due to shear. Thus, membrane emulsification can be used to treat sensitive molecules.

Microchannel emulsification is a similar process to membrane emulsification (Figure 8), but dispersed phase passage does not occur through a membrane, but instead through microchannels fabricated by photolithography.

The main advantages of these methods are low energy consumption and droplet formation with a narrow size distribution, which limits Ostwald ripening and thus increases emulsion stability. However, droplet size is generally larger than with high-energy emulsification methods, and production rate is much lower [40,46,47,48].

##### Other Low-Energy Emulsification Methods

Other low-energy emulsification methods can be used, such as phase inversion or spontaneous emulsification. However, these methods are specific, require a lot of emulsifiers (or even emulsifier and co-emulsifier combination), and cannot be used for large scale production [45].

Phase inversion corresponds to emulsion structure inversion: continuous phase becomes dispersed phase and conversely. This inversion is due to the modification of various parameters such as temperature, electrolytes concentration, oil/water ratio, or even pressure. This method enables to produce concentrated and monodisperse emulsions [38,46].

Spontaneous emulsification occurs when interfacial tension between two immiscible liquids is very low. A weak energy input can enable to accelerate emulsification [46]. A specific case of spontaneous emulsification without emulsifier or mechanical agitation, called Ouzo effect, can be observed as result of change in the components’ proportion leading to oil solubility modification. For example, O/W emulsion can be formed by adding water to oil and solvent solution [49].

### 2.2. Specific Case of High-Frequency Ultrasounds Process

#### 2.2.1. Definition and Generalities

High-frequency ultrasounds correspond to a frequency range greater than 100 kHz and have a low energy, below 1 W/cm^2^ [41]. In this ultrasound range, cavitation phenomena are less important, limiting bubbles collapse consequences. Indeed, when ultrasound frequency increases, cavitation bubbles size decreases because high frequencies reduce time and space for bubbles’ growth. Thus, since cavitation bubbles are smaller, energy released when they implode is weaker [50,51].

#### 2.2.2. Emulsification and Stabilization by High-Frequency Ultrasounds

Emulsification by high-frequency ultrasounds is a method used to produce emulsifier-free emulsions. Two types of high-frequency ultrasounds emulsification can be used: (i) tandem treatment consisting in sequentially irradiating oil/water mixes with different frequencies ultrasounds, ranging from the lowest to the highest frequencies (Table 1) and (ii) single treatment performed with ultrasounds at a specific high-frequency (Table 2).

Kamogawa et al. [52] were one of the first authors to use tandem emulsification. They produced oleic acid/water emulsion by low-frequency ultrasounds (40 kHz for 8 min), followed by high-frequency ultrasounds (0.2 MHz or 0.2 MHz/1 MHz, for 8 min). They showed that mean droplet size was smaller when emulsion was treated by tandem rather than by single low frequency ultrasounds (100 nm and 140 nm for both tandem treatments and 232 nm for single treatment, respectively). In addition, droplet monodispersity and emulsion stability were improved when emulsions were treated by tandem emulsification. Moreover, the authors showed that single high-frequency treatment was not sufficient to emulsify oleic acid/water mix. Indeed, they explained that high-frequency ultrasounds generated weak cavitation and then did not reduce droplet size sufficiently to stabilize emulsion. Single high ultrasound treatment, at 1 MHz, permitted to emulsify only 4% (*w/w*) oleic acid, whereas up to 80% (*w/w*) were incorporated with tandem treatment (40 kHz−0.2 MHz). Thus, high-frequency ultrasounds should be used only to reduce and stabilize pre-formed droplets [52].

Yasuda et al. [55] also formulated oleic acid/water emulsions by tandem emulsification, but using five frequencies: 20 kHz, 0.5 MHz, 1.6 MHz, 2.4 MHz, and 4.8 MHz. Droplet size decreased and emulsion stability increased with increasing sequential treatment. However, oil volume fraction had an impact on emulsification and stability. Emulsions produced sequentially with the five frequencies and with oil volume fraction of 8.0 × 10^−4^ (0.08%) were still stable after 7 months, whereas if oil volume fraction increased, up to 0.03 (3%), emulsions were less stable. This was probably due to insufficient reduction in droplet size and decrease in distance between droplets for higher oil volume fraction [55].

Nakabayashi et al. [53] prepared 3,4-ethylenedioxythiophene/water emulsion by tandem treatment with 20 kHz, 1.6 MHz, and 2.4 MHz ultrasounds. After 5 min of each treatment, they obtained transparent emulsion with droplets smaller than 100 nm. In addition, during this sequential ultrasounds treatment from low to high ultrasounds frequencies (20 kHz; 1.6 MHz; 2.4 MHz), ζ-potential decreased regularly (−29 mV; −37 mV; and −42 mV, respectively). This would be due to OH^−^ adsorption at the oil/water interface [53].

The works cited above concerned emulsions with low oil volume fraction. Kaci et al. [20] studied vegetable oil/water emulsions with higher oil volume fractions (5–15%) treated by single high-frequency ultrasounds (1.7 MHz), without pre-emulsification, and longer treatment time, up to 10 hours. In these conditions, high-frequency ultrasounds could emulsify oil/water mix, but droplet size was large, about 1 µm [20], compared to nanosized droplet obtained by tandem emulsification (Table 1). The pre-emulsification by high-speed homogenization followed by high-frequency ultrasounds treatment (1.7 MHz for 1 h) gave droplet size much smaller, between 150 and 250 nm, but still with the presence of larger droplets, approximately 2 µm [62,63]. When emulsions were formulated with 5% sunflower oil and treated by high-frequency ultrasounds at 1.7 MHz, better stability was obtained than emulsions formulated by low-frequency ultrasounds (40 kHz) or by HPH, at 1500 bar. The improvement of emulsion stability treated by high-frequency ultrasounds could be due to a majored population of nanoscale sized droplets compared to low-frequency ultrasounds or HPH emulsions [63]. In addition to breaking up droplets, high-frequency ultrasounds would make droplets surface rather hydrophilic, which would also participate in their stabilization [52].

In addition to the O/W emulsions presented above, W/O emulsions can also be obtained by high-frequency ultrasounds. Nakabayashi et al. [57] produced W/O emulsion with potassium carbonate solution as dispersed phase (10%—*v/v*) and chloroform as continuous phase. Treating emulsions by tandem emulsification (20 kHz, 1.6 MHz, and 2.4 MHz ultrasounds) gave small droplets of few hundred nanometers. It was shown that droplet size was smaller when this tandem emulsification was performed twice (112 nm) than once (436 nm). W/O emulsions formulated by two tandem ultrasounds cycles were stable for more than 6 months [57].

#### 2.2.3. High-Frequency Ultrasounds Emulsification Uses

Emulsions formulated by high-frequency ultrasounds are stable without emulsifier, which could allow to use them in numerous potential applications.

In the chemistry field, emulsifier-free emulsions formulated by high-frequency ultrasounds can be used to synthetize polymer nanoparticles [56,58,59], to produce conductive polymer films or coatings [53,54] or to perform chemical reactions [57,60].

Emulsifier-free emulsions formulated by high-frequency ultrasounds can also be used in the cosmetic and pharmaceutic fields to encapsulate and vectorize biomolecules. For example, when Coenzym-Q10 is vectorized into emulsifier-free emulsion and treated by high-frequency ultrasounds, it exhibits better activity than when it is vectorized into emulsion containing emulsifiers and formulated by traditional methods such as low-frequency ultrasounds or HPH [62]. Another example is biomolecules delivery, such as caffeine. When this biomolecule was introduced to emulsions and treated by high-frequency ultrasounds, its diffusion rate into skin was not affected by treatment, compared to reference emulsion formulated by low-frequency ultrasounds and containing emulsifiers. This study showed that high-frequency ultrasounds treatment can decrease active ingredients content in cosmetic formulations without affecting their effectiveness [64].

Thus, the high-frequency ultrasounds process is an emulsification method with potential opportunities in many areas, such as cosmetic and pharmaceutical industries, but also in the synthetic chemistry field.

#### 2.2.4. High-Frequency Ultrasounds Emulsification Drawbacks

Despite its promising applications, high-frequency ultrasounds use could cause damage to treated products.

It has been shown that high-frequency ultrasounds could produce free radicals such as hydroxyl radical (OH^•^) [65] or singlet oxygen (^1^O_2_) [66]. OH^•^ oxidizes lipids by a chain mechanism, called auto-oxidation, which consists of three steps: initiation, propagation, and termination. During this mechanism, one radical can oxidize hundreds of lipids. ^1^O_2_ is a reactive oxygen species which can oxidize lipids by a non-radical mechanism. ^1^O_2_ does not react with double bound by a chain mechanism, but by a stoichiometric reaction [67]. Thus, OH^•^ and ^1^O_2_ presence in emulsions can cause lipid degradation and thus impacts products’ chemical, physical, and sensory properties, affecting their quality and safety [68,69]. It was shown, by Fourier transform infrared spectroscopy (FTIR) measurements, that oil treated by high-frequency ultrasounds was not degraded after treatment. However, after 30 days of storage at 37 °C, oil was oxidized [63]. These results confirmed that oil was not degraded during emulsification, but this did not exclude reactive oxygen species production during treatment, which then oxidized oil during storage.

Another drawback of high-frequency treatment is a possible phase separation during storage. Indeed, experimental and modeling studies showed that ultrasounds between 1 and 2 MHz could favor creaming and droplets coalescence [70,71,72,73,74]. This coalescence would be due to secondary acoustic forces which induce attraction between droplets [75]. Difference between droplets formation and droplets coalescence could come from power and/or time of treatment.

### 2.3. Conclusion on Emulsification Processes

Emulsification can be carried out by various processes. Each process presents advantages and drawbacks, and depending on applications, not all processes can be used.

A recent process, using high-frequency ultrasounds, seems promising and does not require stabilizer to obtain stable emulsion, which is a significant advantage, in a context where manufacturers try to reduce additives from their products. However, high-frequency ultrasounds’ effect on emulsions stabilization is not yet explained. Thus, this process requires more research to better understand its effects and the mechanisms by which stabilization is obtained. This method allows to produce emulsions in the absence of any kind of stabilizer. The next part of the present review consists in understanding how the emulsifier-free interface is organized in order to try to explain high-frequency ultrasounds’ effect on the interface.

## 3. Emulsifier-Free Oil/Water Interface Organization

An interface is defined as “a narrow region that separates two phases, which could be a gas and a liquid, a gas and a solid, two immiscible liquids, a liquid and a solid or two solids. The two phases may consist of different kinds of molecules (e.g., oil and water) or different physical states of the same kind of molecule (e.g., liquid water and solid ice).” [76]. Based on surface charge and ζ-potential studies (see Section 3.1.1), the oil/water and air/water interfaces are comparable because water behaves similarly with low dielectric hydrophobic surfaces, such as oil and air [25,77,78,79]. Studies bearing on the oil/water interface and on the air/water interface are presented here to try to understand the interfacial structure of emulsifier-free emulsions because of the lack of results on the interfaces of emulsion O/W and W/O.

The first studies about the structure of emulsifier-free hydrophobic/water interfaces, also called “pristine hydrophobic/water interfaces” began more than 150 years ago with air/water interface studies. In 1861, Quincke showed by electrophoresis measurement that air bubbles were negatively charged due to their migration toward the positive electrode [80]. Oil/water interface studies are more recent. In 1938, Carruthers was one of the first to demonstrate that oil droplets were also negatively charged [81].

Since then, many authors tried to understand the hydrophobic/water interface structural organization thanks to different macroscopic and microscopic measurement methods, but also thanks to simulation methods. Some of them confirmed that the interfaces were negatively charged and would be basically due to hydroxide ions adsorption at the interface [23,24,25], but others showed that the hydrophobic/water interface would be rather acid due to hydronium ions’ position at the interface [28,29] (Table 3).

### 3.1. First Hypothesis: Hydroxide Ions Adsorption at Interface

#### 3.1.1. Macroscopic Measurements

The first macroscopic analyses to study hydrophobic interfaces concerned electrophoretic mobilities and ζ-potential measurements to determine droplets surface charge. Thus, numerous authors showed that oil droplets, or air bubbles, dispersed in the aqueous phase had a negative surface charge [21,22,23,24,25,77,81,84,95,96].

These authors showed a high dependence between surface charge and pH. For example, Carruthers [81] studied the electrophoretic mobility of different organic substances (*n*-octadecane, octadecene, undecyl and octyl alcohols, halogenated octanol derivatives, as well as ethyl laurate) in the form of droplets dispersed in water containing 0.01 M NaCl for pH ranging from 2 to 12. The droplets were positively charged as they migrated towards the cathode, at pH below 2.5–3. Above these pH values, droplets were negatively charged and electrophoretic mobility towards the anode increased until a maximal value at pH 8–10 [81]. Similarly, strong dependence between ζ-potential and pH were obtained with other oils and alkanes, such as xylene, dodecane, hexadecane, or perfluoromethyldecalin [23,84]. For alkanes, it was demonstrated that ζ-potential were chain length-dependent. For alkanes containing 6, 7, or 8 carbon atoms, the ζ-potential-zeta decrease with pH increase from 6.5 to 11 was twice less important than with alkanes containing 9 to 16 carbon atoms. This difference might be explained by the higher solubility in water of short alkanes [84].

Due to the strong dependence between electrophoretic mobility (or ζ-potential) and pH, Dickinson [21] made the hypothesis that the origin of droplets’ negative charge would be due to OH^−^ adsorption at the interface. This hypothesis was approved by Beattie’s team using pH-stat experiments, who showed that hexadecane/water emulsion pH decreased during emulsification. The authors added hydroxide ions to maintain a constant pH value. Moreover, they observed a linear correlation between the amount of hydroxide ion added and the increase of surface area created during emulsification. Thus, they concluded that OH^−^ adsorbed at the interface stabilized emulsions. From these experiments, they determined that surface-charge density at the interface was comprised from −5 to −7 µC/cm^2^, which corresponds to 0.31 unit charges per 100 Å^2^. In other words, at the oil/water interface, one OH^−^ would be present on every 3 nm^2^ [24,25,82].

Dickinson proposed two possible mechanisms to explain hydroxide ions adsorption at the interface: (i) OH^−^ adsorption from the aqueous phase depends on its pH value, and reaches a maximum at pH 9, due to the saturation of the interfacial surface; (ii) adsorption of a unimolecular layer of water molecules at the oil surface. In this last hypothesis, some of the water molecules at the interface would ionize leading to the partition of hydronium ions towards the bulk, while hydroxyl ions would adsorb at the interface. This ionization would depend on bulk pH: when pH increases, ionization increases too, thus, the number of OH^−^ adsorbed increases. Dickinson proposed that those mechanisms could take place individually or in combination [21]. More recently, researchers supposed water ionization at the interface would be due to electric field gradient presence, via the second Wien effect. Indeed, the specific orientation of H_2_O molecules, where hydrogen atoms point towards the phase with lower dielectric constant, would generate an electric field, favoring water autolysis (Equation (4)):(4)$${2\;H}_{2}O\;↔{\;H}_{3}O^{+}{+\mathrm{OH}}^{-}$$

The water dissociation constant (pK_w_) at the droplet surface is about 8, against 14 in bulk. Thus, water autodissociation would be 10^6^ times more important at the surface, than in bulk. On the other hand, electric field gradient at the interface would favor OH^−^ adsorption and protons repulsion [78,82,91,95,124,125,126].

After Dickinson, other authors provided more details about the mechanism explaining why OH^−^ are adsorbed at the interface. This phenomenon would be due to the fact that hydroxide ions are able to reduce water dielectric constant and dipolar fluctuations with bulk H_2_O molecules, causing Hamaker-like force and thus attracting OH^−^ at the interface, where dipole moment fluctuations are lower than in bulk water [77,109].

#### 3.1.2. Spectroscopic and Simulation Studies on Interfaces

Only few methods at the microscopic scale allow to study interface organization due to the difficulty to distinguish the ionic species present at the interface and those in the bulk phase. However, two second order nonlinear spectroscopic techniques (sum frequency generation (SFG) and second-harmonic generation (SHG) methods) can probe the interfacial region [106,127]. These spectroscopic methods use two pulsed laser beams which interact coherently in space and time. Emitted photons own a frequency (ꙍ_SFG_ or ꙍ_SHG_) corresponding to the sum of the two incident frequencies. SFG spectroscopy uses an infrared (IR) beam and an UV/Visible (UV/Vis) beam, thus ꙍ_SFG_ = ꙍ_IR_ + ꙍ_UV/Vis_, whereas for SHG spectroscopy, the two beams are in the UV/Visible domain, and so ꙍ_SHG_ = 2ꙍ_UV/Vis_ [128,129,130].

SFG coupled with isotopic dilution experiments showed emerging hydroxide ions at the interface, but this phenomenon was only observed at pH above 13 [102]. Similarly, Tian’s team used phase-sensitive SFG method and showed that at the water/vapor interface, OH^−^ accumulated in 1.2 M NaOH solution [106]. Even if no signal proved OH^−^ adsorption at the interface, in pH range 2–13, that does not necessarily mean that OH^−^ does not adsorb at the interface, but just that its spectroscopic signal is too weak to be observed [77]. Regarding the oil/water interface, phase-sensitive SFG showed, at pH above 3, that OH^−^ accumulated at the octadecyltrichlorosilane/water interface [88] and at the hexadecanol/water interface [89].

Second harmonic generation spectroscopy also showed that OH^−^ accumulate at the hexadecane/water interface and that OH^−^ adsorption depended on pH (the higher pH, the higher second harmonic signal intensity). Moreover, it has been shown that negative charges at the interface could not be due to impurities, because signal corresponding to OH^−^ accumulation was the same in the presence or absence of impurities [26]. Another study showed that hydroxide ions adsorption at the hexadecane/water interface saturated at a relatively low alkali concentration (almost 1 mM), which consolidates the hydroxide ions specific affinity at the oil/water interface hypothesis [87].

OH^−^ ions adsorption at the air/water interface was also shown by fluorescent spectroscopy with specific fluorescent dyes use, i.e., *n*-decylfluorescein and *n*-decyleosin. After demonstrating that dyes were most likely located at the air/water interface, the authors showed that above pH 3, dyes were negatively charged, which demonstrated OH^−^ are adsorbed at the interface [99].

Several simulation studies alone or combined with experimental studies showed that hydroxide ions are adsorbed at the hydrophobic medium/water interface [101,107,111,112,121]. According to quantum mechanical calculations, gaseous carboxylic acids deprotonation at the interface would be inhibited by a kinetic barrier, unless OH^−^ would be present. Thus, by demonstrating deprotonated acids’ presence at the interface by mass spectrometry, Mishra et al. proposed the OH^−^ presence at the hydrophobic interface. Moreover, they showed that isoelectric point was near pH 3, which is coherent with oil droplet and air bubbles ζ-potential measurements [111].

Recently, the combination of an experimental study by SFG and a theoretical study by ab initio molecular dynamics (AIMD) simulations confirmed hydroxide ions accumulate at the hexane gas/water interface. SFG measurements had even enabled to link hexane coverage and OH^−^ surface density, where five adsorbed hexane molecules adsorbed one OH^−^. AIMD simulations showed that adsorbed hexane molecules formed an interfacial structure with an H-bonding network of water, which decreased OH^−^ mobility and suppressed OH^−^ diffusion from the interface to the bulk so that OH^−^ accumulated at the interface. The authors suggested that the hydrophobic hydrocarbons/water interfaces are basic [101].

Recent calculation studies of adsorption energies of OH^−^ and H_3_O^+^ at the air/water interface showed that OH^−^ adsorption is energetically favorable, whereas H_3_O^+^ adsorption is unfavorable. HCO_3_^−^ adsorption would be also favorable, but to a lesser extent than OH^−^ [112].

All these studies tend to validate the OH^−^ adsorption phenomenon at the interface.

#### 3.1.3. Other Origins of the Negative Charge

Other hypotheses about negative charge of droplets or bubbles were tested. First, authors thought anions, other than OH^−^, adsorbed at the interface. Marinova et al. showed that Cl^−^ adsorption at the interface cannot explain ζ-potential results obtained, thus reinforcing the OH^−^ ions adsorption hypothesis [23]. Beattie’s group showed that the addition of Cl^−^, I^−^, ClO_4_^−^, or dipolar anions, such as SCN^−^, IO_3_^−^ and CH_3_COO^−^, in hexadecane/water emulsions did not impact the dependence between ζ-potential and pH. Thus, these experiments showed that preferential OH^−^ adsorption at the interface was not due to dipole-dipole forces nor electrostatic attraction, but was specific to hydroxide ions [24,82]. In hexadecane/water emulsions, Franks et al. [83] added different anions (Cl^−^, Br^−^, I^−^, F^−^, ClO_4_^−^, or IO_3_^−^) that have different polarizability and, according to lyotropic series, different hydration enthalpy. The authors showed that neither hydration enthalpy nor polarizability had an effect on the ζ-potential of hexadecane droplets in water. They then concluded that another property of hydroxide ions would be at the origin of their strong adsorption at the interface and thus supposed that it would be due to hydrogen-bonding interactions with water molecules at the interface [83]. A more recent study showed that OH^−^ would not be the only ion to accumulate at the interface. Indeed, according to phase-sensitive SFG measurements, other ions, such as I^−^, Cl^−^, H_3_O^+^, as well as Na^+^, could accumulate at the alcohol/water interface, but OH^−^ was the ion with highest affinity [89].

Another hypothesis about the negative charge of droplets and bubbles would be negative adsorption or depletion, of cations, mostly hydronium ions. However, some authors showed that this hypothesis was unrealistic by calculating the depletion layer thickness [23], the hydronium and hydroxide concentrations [125], as well as surface potential [131].

To explain why negative charge is found at the interfaces, some authors proposed that this would be due to HCO_3_^−^ and CO_3_^2−^ ions coming from air CO_2_ dissolution [91,94]. Experiments realized under N_2_ with various content of Na_2_CO_3_ were not conclusive [23,24]. However, recently, by using the modified Poisson-Boltzmann equation, Iyota and Krastev showed that negative surface charge would not only be due to hydroxide ions, but also due to CO_2_ derived ions, such as HCO_3_^−^ and CO_3_^2−^, and also Cl^−^ [100].

Negative charge could be due to the long-range orientation of water dipoles at the interface, but experiments using chaotropic agent urea, that can destroy molecular network in the bulk, showed electrostatic potential did not change in absence or presence of urea [23].

Recently, other hypotheses about negative charge surface were analyzed. State-of-the-art linear scaling density functional theory use showed negative charge would be due to charge transfer, and not negative ions adsorption. Indeed, at the oil/water interface, a charge transfer from water molecules to oil molecules would cause a negative charge of these latter molecules [93]. Similar results were obtained for the air/liquid [122] and solid/liquid interfaces. In this last case, a hybrid model, called Wang’s hybrid electric double layer model, was developed. A first step would consist of electron transfer at the surface, followed by a second step where free ions were attracted to the electrified solid surface. Simultaneously, water ionization reactions should occur at the surface. After losing an electron, H_2_O molecule becomes a water cation H_2_O^+^ with a very short lifetime and combines rapidly with another H_2_O molecule according to Equation (5), where ion and radical formed would enable electric double layer formation:(5)$$H_{2}O^{+}{+H}_{2}O\;\to {\;H}_{3}O^{+}+\mathrm{OH}.$$

This hybrid model described for the solid/liquid interface might also correspond to the liquid/liquid interface [132,133].

### 3.2. Second Hypothesis: Hydronium Ions Adsorption at Interface

Surface tension measurement is an indirect method to determine physical ion adsorption at the interfaces. According to Gibbs adsorption equation, ions decreasing surface tension are attracted toward the interface, whereas ions increasing surface tension are repelled from the interface. Thus, according to surface tension measurements from aqueous electrolyte solution, H_3_O^+^ would be adsorbed at the interface, whereas OH^−^ would be repelled from the interface [28,29,30,134]. However, these conclusions could be invalid because Gibbs isotherm used did not consider cation number changes. When these modifications are considered, OH^−^ ions adsorption at the interface led to surface tension almost independent of pH, for pH values comprising between 1 and 13, unless other cations than H_3_O^+^ were present. Thus, by applying the corrected Gibbs isotherm, experimental surface tension data showed that OH^−^ ions adsorbed at the interface, and more precisely, about 0.7–1 nm below the surface [108,123].

Another study, combining surface tension measurements and a thermodynamic model-based Gibbs concept showed that OH^−^ and H_3_O^+^ had a greater affinity for the interface than for the bulk, but OH^−^ affinity would be more important than that of H_3_O^+^ [120].

Several spectroscopic methods used to study and understand interfacial organization showed that hydroxide ions would not be adsorbed at the interface, but that it would rather be hydronium ions.

SFG spectroscopy [103] combined with isotopic dilution experiments [102] brought out that hydronium ions, Zundel (H_5_O_2_^+^) and Eigen (H_3_O^+•^(H_2_O)_3_) forms, could be adsorbed at the air/water interface. It is important to note that, Eigen form was observed only in SFG spectroscopy combined with isotopic dilution experiments [102]. However, all these observations were only made at low pH (below pH 2), which were coherent with ζ-potential measurements defining an isoelectric point around pH 2–4 [25]. More recently, phase sensitive SFG spectroscopy was used to demonstrate H_3_O^+^ adsorption at the air/water interface. Authors also brought out differences in pH between the surface and bulk, which was of −0.65 (+/−0.14) pH unit [105]. However, like for previous experiments, this study was conducted in acidic conditions, which does not provide any proof of H_3_O^+^ presence at the interface when pH is superior to the isoelectric point.

By using SHG method, Petersen and Saykally demonstrated an increase H_3_O^+^ concentration at the liquid/water surface, but that was shown by indirect method. They observed an increase iodide concentration at the hydriodic acid solution surface, which would be an indicator of an increased hydronium concentration, according to theoretical studies [27]. Later, these same authors showed, from SHG measurements and Langmuir adsorption model, that hydroxide ions would be repelled from the surface. This study was conducted with basic solutions having a pH range from 9 to 14 (potassium and sodium hydroxide solutions). However, a weak OH^−^ adsorption would not be rejected, due to possible experimental uncertainty [104].

Jungwirth’s team used a linear surface selective spectroscopic method, synchrotron photoelectron spectroscopy (PS), and molecular dynamics simulations to study the interface between vapor and aqueous basic solutions. They showed that OH^−^ were “weakly repelled or at best very weakly attracted to the surface” [29]. An uncertainty therefore seems to exist as to OH^−^ adsorption. According to Beattie, these results would be due to a lack of depth of the spectroscopic probe used which probed only the 1 to 3 first water layers, whereas OH^−^ would be positioned deeper. With a more penetrating probe, more OH^−^ were highlighted. That would explain why these ions are not observed with some spectroscopic methods [135]. However, according to Jungwirth’s team, OH^−^ layer present at one or several nanometers from the surface would be observable if it was well present. As it is not the case, that confirmed OH^−^ would not be adsorbed at the interface [136]. Nevertheless, PS experiments duration would be too short to obtain surface equilibrium. That is another experimental bias explaining why OH^−^ would not be observable [135].

Simulation studies showed also that hydronium ions would be adsorbed at the interface [28,113] and hydroxide ions would be even repelled from the interface [30,115,116,117]. The interface would be thus acidic with a surface pH value of 4.8 [114]. However, using first-principles molecular dynamics simulations, Mundy et al. showed that H_3_O^+^ and H_5_O_2_^+^ ions adsorption would be less important than predicted previously by Buch et al. [114] and it might even be OH^−^ presence at the air/water interface [110]. Later, this same team’s research suggested that both hydronium and hydroxide ions were present at the air/water interface [119].

### 3.3. Other Hypotaheses

#### 3.3.1. No Charge at Interface

Some authors thought the interface would be not charged [85,92,137]. Thus, negative ζ-potential would be due to stabilizer impurities presence, but this hypothesis was ruled out by purifying several times materials used to formulate emulsions, verifying oil/water interfacial tension, studying oil purity by UV and IR spectroscopy measurements, and analyzing results reproducibility, which could not have been obtained with impurities noncontrolled traces [23]. However, Roger and Cabane thought that negative charge would be due to fatty acid accumulation at the oil/water interface, which have stabilizer properties. Thus, OH^−^ added during emulsification would react with fatty acids, and not adsorbed at the interface [86,137]. Nevertheless, the comparison between emulsions formulated with purified or unpurified oils showed it was necessary to add more hydroxide ions to maintain pH during emulsification with unpurified oil than with purified oil. That confirmed OH^−^ reacted with fatty acids, but it was always necessary to add OH^−^ during emulsification with purified oil, confirming OH^−^ adsorbed at the interface [138]. On the other hand, spectroscopic measurements confirmed carboxylic acid absence at the interface and thus surface charge would be due to oil/water interface property [139].

A molecular dynamics simulation study reproduced in the presence of an electric field, without any ion. This study showed that uncharged oil droplets dispersed in water can have negative electrophoretic mobility, and thus mobility electrophoretic would not reflect the charge of oil droplets [92]. However, it was then shown, by calculation, that oil droplets, and more widely hydrophobic particles, had negative electrophoretic mobility only when they were negatively charged [140].

More recently, non-charged emulsions would have been formulated by using rigorous cleaning procedures and specific solvent storage conditions. SFG spectroscopic study confirmed clean interface formation. These emulsions were stable during several days [85]. However, it was then shown that uncharged *n*-hexadecane droplets [141] and uncharged bubbles [142] dispersed in water were unstable. Indeed, electrophoretic mobility must be sufficient to avoid droplets coalescence, thanks to electrostatic repulsions. Moreover, specific cleaning procedures would not have any effect on oil droplets ζ-potential [141].

#### 3.3.2. H_3_O^+^ and OH^−^ Presence

Numerous authors thought there is not only OH^−^, nor only H_3_O^+^, but both ions at the interface to ensure electroneutrality of the double layer [25,77,98,112,119,121,123]. Two main hypotheses appeared around this supposition: (i) H_3_O^+^ would be placed in the surface layer, although OH^−^ would be in the diffuse layer (Figure 9A), according to simulations and spectroscopic studies [28,118]. (ii) OH^−^ ions would be adsorbed in the surface layer, whereas H_3_O^+^ would be placed in the diffuse layer (Figure 9B). OH^−^ contributes more to the surface charge than H_3_O^+^ because ζ-potential is measured at 2–3 nm depth and its value reflects the charge between shear plane and surface [25,109].

### 3.4. Conclusion on Interfacial Organization

To synthesize, the structure and organization of the hydrophobic/water interfaces were the subject of a long debate and despite numerous studies full agreement is still not reached. Indeed, since 2005, evaluation methods have increased, but no consensus was reached, and several hypotheses were formulated (Figure 10). The two main hypotheses are hydroxide or hydronium ions adsorption. However, other ions seem to be adsorbed at the interface, with less affinity than the two first cited. Thus, interfacial structure is very complex, which can explain study difficulty and disagreement between authors.

On the one hand, according to macroscopic studies, the water surface layer is negatively charged due to hydroxide ions adsorption. However, recently, some studies showed that the negative charge origin would not be clear because of possible charge transfer taking place alone, or in combination with OH^−^ adsorption.

On the other hand, numerous simulation and spectroscopy studies are in conflict with macroscopic measurements. The hydrophobic/water interface would be stabilized by hydronium ions. However, some recent spectroscopic measurements are coherent with macroscopic studies, which could be due to the improvement of spectroscopic techniques. According to Petersen et al., SHG and SFG probing depths were not well founded [104] and it was shown that OH^−^ gradients would be present at the interface, explaining why spectroscopic studies revealed only a weak signal corresponding to OH^−^ adsorption [111]. Thus, probing methods were improved which could explain why more recent spectroscopic studies demonstrated that hydroxide ions were preferentially adsorbed at the interface. Moreover, numerous recent studies showed that H_3_O^+^ and other ions would also be present at the interface, but in the diffuse layer to respect electroneutrality.

It is also important to remark that most of studies showing that H_3_O^+^ is adsorbed at the interface have been conducted on the air/water interface (Table 3). Thus, it seems that contrary to what was stated at the beginning of Section 3, the air/water and oil/water interfaces are not totally comparable, especially when microscopic structure is studied. Indeed, recently, some works showed that water molecules did not behave in the same way at the air/water or oil/water interfaces [89,143,144], which could explain different results obtained and different hypotheses formulated on water interfacial organization.

Thus, despite new approaches and more developed methods use, numerous controversies still exist about the charge and organization of hydrophobic/water interfaces, but it seems that the interface is more complex than that defined in the early 2000s. It would consist of several ion types organized in different layers with other possible mechanisms, such as surface charge transfer.

Although oil/water interfacial organization in the absence of any kind of stabilizer is not yet well understood, the elements presented in this part could help to understand the effect of high-frequency ultrasounds on the oil/water interface and thus on the stabilization of emulsifier-free emulsions.

## 4. Conclusions

Emulsions are metastable systems used in various fields. Numerous methods enable to formulate emulsions, such as low-frequency ultrasounds, HPH, microfluidization, membrane and microchannel emulsification, as well as spontaneous emulsification and phase inversion. With these different processes, emulsifiers are generally necessary to stabilize emulsions over time.

However, a new process using high-frequency ultrasounds can stabilize emulsions without emulsifier, and vectorize/encapsulate biomolecules, and is thus promising in food and cosmetics fields. This process also makes it possible to reduce droplet size, and even to emulsify oil/water mix. The stabilizing effect is currently unknown and requires further studies. To try to understand high-frequency effect on the interfaces, emulsifier-free interface organization is an important way to study, which was the objective of the third part of this review.

Emulsifier-free interfacial organization was studied by various macroscopic and spectroscopic methods, but also by simulation studies. Despite these numerous studies, no consensus has been found by authors and numerous discords exist between research teams. Macroscopic studies consisting essentially of measuring droplets surface charge and some spectroscopic and simulation studies showed that OH^−^ would adsorb at the interface, whereas most spectroscopic and simulation studies suggest that H_3_O^+^ would be preferentially adsorbed at the interface. Thus, more studies would be necessary to better define emulsifier-free interface organization. However, in view of all the elements presented in this review, we suppose that the oil/water interface is negatively charged with the majority presence of OH^−^ ions. We also assume that other ions would be present at the interface, but in less quantity.

Thus, due to the complexity of the oil/water interfacial organization and to the difficulties in studying and defining this organization, it could be difficult to understand high-frequency ultrasounds effect on the interface, but perhaps high-frequency ultrasounds effect study on the interface could, on the contrary, provide new evidence to better understand emulsifier-free interfacial organization.

The effect of high-frequency ultrasounds on biomolecules encapsulation and delivery also require more research. This could then allow to optimize this process for food applications, such as in the formulation of functional ingredients or foods. An immediate industrial application could be focused on food supplements. High-frequency ultrasounds enable to obtain emulsions highly enriched in both hydrophilic and hydrophobic nutrients (i.e., vitamins, essential oils, polyunsaturated fatty acids) without any additives, such as emulsifiers or other stabilizers. This kind of product is an excellent answer to people’s requirements for natural and additive-free food products.

## Figures and Tables

**Figure 1 foods-11-02194-f001:**
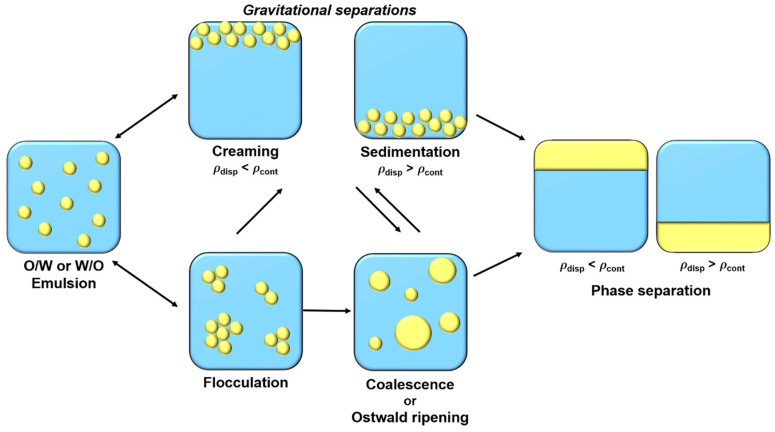
Schematic illustration of emulsions destabilization mechanisms.

**Figure 2 foods-11-02194-f002:**
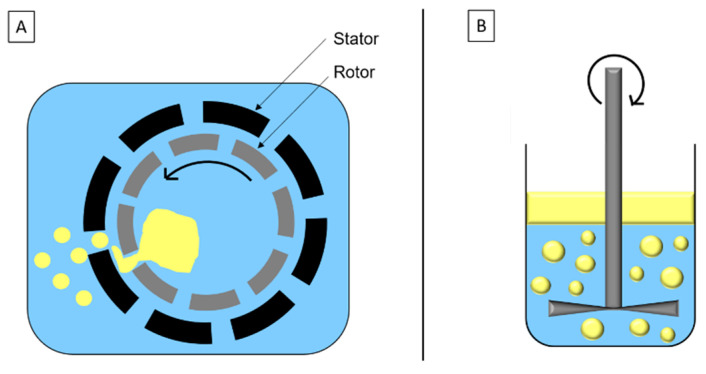
Schematic representation of (**A**) high-speed homogenization and (**B**) disc system.

**Figure 3 foods-11-02194-f003:**
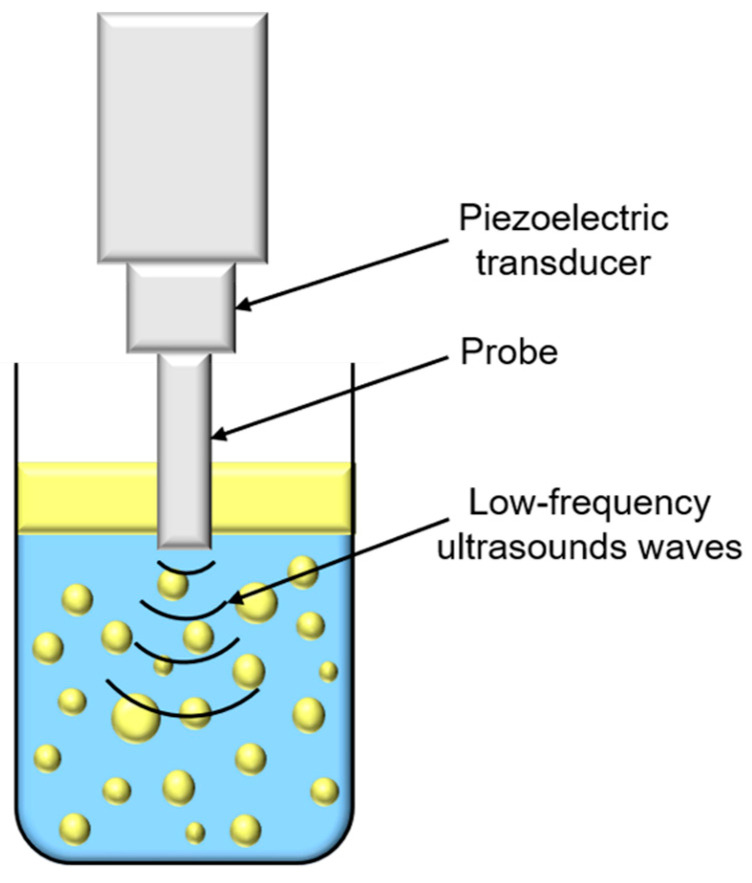
Schematic representation of low-frequency ultrasounds treatment.

**Figure 4 foods-11-02194-f004:**
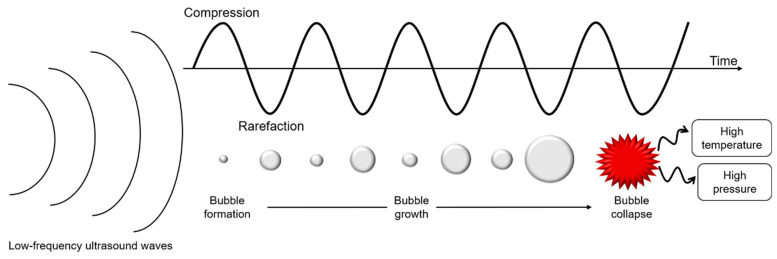
Schematic representation of cavitation phenomenon under low-frequency ultrasounds treatment.

**Figure 5 foods-11-02194-f005:**
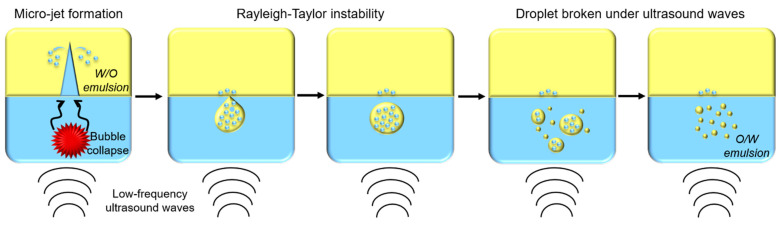
Schematic representation of oil-in-water (O/W) emulsion formation by low-frequency ultrasounds.

**Figure 6 foods-11-02194-f006:**
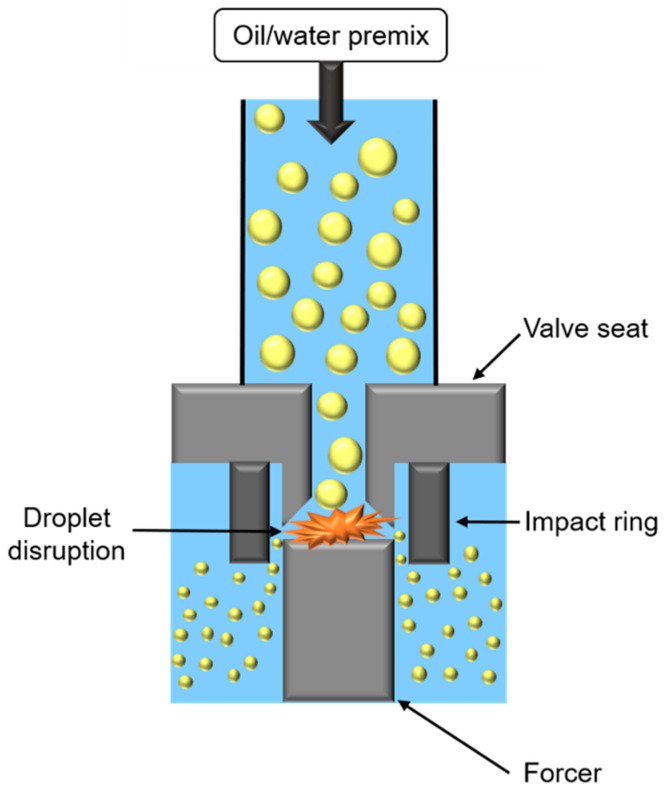
Schematic representation of high-pressure homogenizer.

**Figure 7 foods-11-02194-f007:**
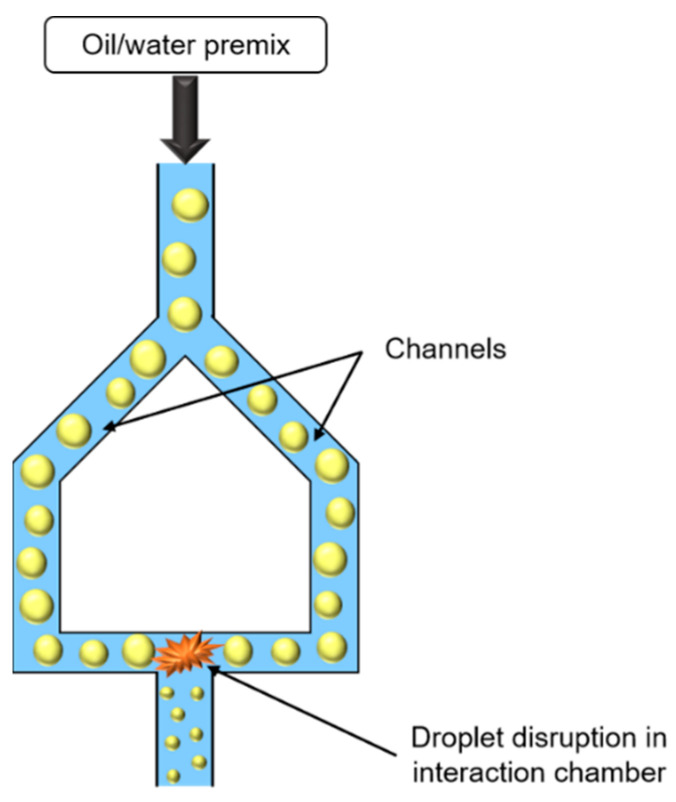
Schematic representation of microfluidizer.

**Figure 8 foods-11-02194-f008:**
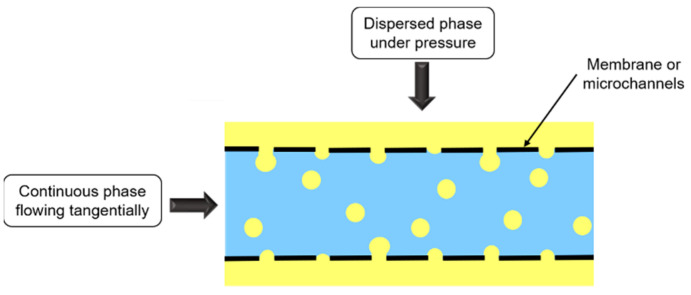
Schematic representation of membrane and microchannel emulsifications.

**Figure 9 foods-11-02194-f009:**
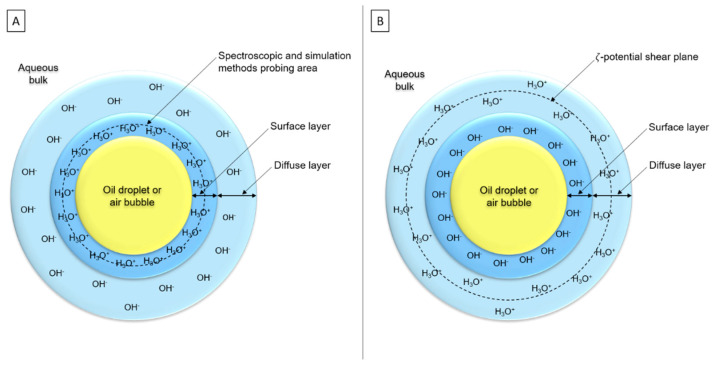
Schematic representations of the two main hypotheses about interfacial organization: (**A**) H_3_O^+^ adsorption at interface with OH^-^ presence in subsurface layer and (**B**) OH^-^ adsorption at interface with H_3_O^+^ presence in diffuse layer (simplified representations—scales are not respected).3.4. Conclusion on Interfacial Organization.

**Figure 10 foods-11-02194-f010:**
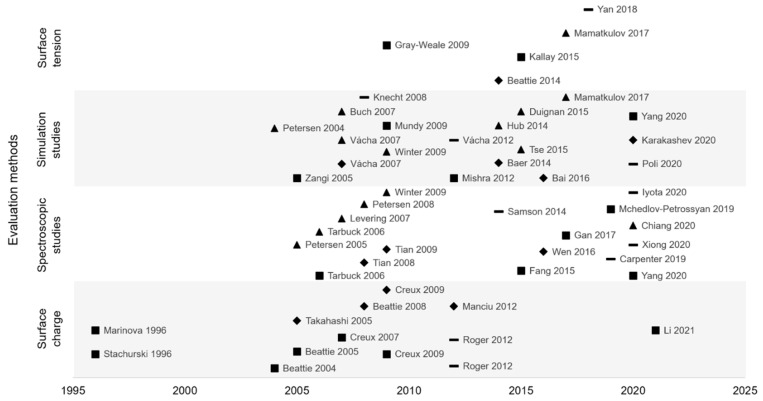
Interfacial organization hypotheses (■ hydroxide ions (OH^−^) adsorption, ▲ hydronium ions (H_3_O^+^) adsorption, ◆ OH^−^ and H_3_O^+^ adsorption, **–** other hypotheses) according to experimental methods used and publication date of papers.

**Table 1 foods-11-02194-t001:** Characteristics of emulsions produced by tandem high-frequency ultrasounds treatment.

Oil (Concentration)	Ultrasounds Frequencies (Treatment Time)	Mean Droplet Size (in nm)	References
Oleic acid (1.4% *w/v*)	40 kHz (8 min)	232	[52]
	200 kHz (8 min)	∼100	
	1 MHz (8 min)	∼350	
	40 kHz (8 min) + 200 kHz (8 min)	∼100	
	40 kHz (8 min) + 200 kHz (8 min) + 1 MHz (8 min)	140	
3,4-Ethylenedioxythiophene (0.3% *w/v*)	20 kHz (5 min)	351	[19,53,54]
	20 kHz (5 min) + 1.6 MHz (5 min)	208	
	20 kHz (5 min) + 1.6 MHz (5 min) + 2.4 MHz (5 min)	82	
Oleic acid (Volume fraction: 8.0 × 10^−4^)	20 kHz (1 min)	∼100	[55]
	20 kHz (1 min) + 0.5 MHz (3 min)	∼90	
	20 kHz (1 min) + 0.5 MHz (3 min) + 1.6 MHz (3 min)	∼70	
	20 kHz (1 min) + 0.5 MHz (3 min) + 1.6 MHz (3 min) + 2.4 MHz (3 min)		
	20 kHz (1 min) + 0.5 MHZ (3 min) + 1.6 MHz (3 min) + 2.4 MHz (3 min) + 4.8 MHz (3 min)		
Oleic acid (Volume fraction: 6.0 × 10^−3^)		∼110	
Oleic acid (Volume fraction: 3.0 × 10^−2^)		∼120	
Chloroform (Volume fraction: 2.0 × 10^−2^)	20 kHz	20,000	
	20 kHz + 0.5 MHZ	<1000	
	20 kHz + 0.5 MHZ + 1.6 MHz + 2.4 MHz + 4.8 MHz	-	
Methyl methacrylate	20 kHz (8 min)	220	[56]
	20 kHz (8 min) + 500 kHz (10 min)	112	
	20 kHz (8 min) + 500 kHz (10 min) + 1.6 MHz (10 min)	51	
	20 kHz (8 min) + 500 kHz (10 min) + 1.6 MHz (10 min) + 2.4 MHz (10 min)	23	
W/O emulsion: potassium carbonate (10% *v/v*) in chloroform	20 kHz (10 min) + 1.6 MHz (10 min) + 2.4 MHz (10 min)	436	[57]
	Two cycles: 20 kHz (10 min) + 1.6 MHz (10 min) + 2.4 MHz (10 min)	112	
Methyl methacrylate	20 kHz (5 min)	103	[58]
	20 kHz (5 min) + 500 kHz (10 min)	87	
	20 kHz (5 min) + 500 kHz (10 min) + 1.6 MHz (10 min)	61	
	20 kHz (5 min) + 500 kHz (10 min) + 1.6 MHz (10 min) + 2.4 MHz (10 min)	42	
Perfluoromethyl-cyclohexane (2.4% *v/v*)	20 kHz (7 min)	175–311	[59]
	20 kHz (7 min) + 500 kHz (15 min)	224–430	
	20 kHz (7 min) + 500 kHz (15 min) + 1.6 MHz (15 min)	342	
	20 kHz (7 min) + 500 kHz (15 min) + 1.6 MHz (15 min) + 2.4 MHz (15 min)	306	
	20 kHz (7 min) + 500 kHz (15 min) + 1.6 MHz (15 min) + 2.4 MHz (15 min) + 5 MHz (15 min)	158	
Allyltriethylsilane (3.75% *w/v*)	20 kHz (5 min)	1202	[60]
	20 kHz (5 min) + 1.6 MHz (5 min)	132	
	20 kHz (5 min) + 1.6 MHz (5 min) + 2.4 MHz (5 min)	59	

**Table 2 foods-11-02194-t002:** Characteristics of emulsions produced by single high-frequency ultrasounds treatment.

Oil (Concentration)	Pre-Emulsification	Ultrasounds Frequencies and Treatment Time	Mean Droplet Size (in nm)	References
Sunflower oil (5; 10 and 15% *v/v*)	-	1.7 MHz (10 h)	∼1000	[20]
Toluene (1% *v/v*)Emulsifier presence: Tween 20 (0.1% *w/w*)	Low-frequency ultrasounds (20 kHz, 4 min) or high-speed homogenizer (6000 rpm—10 min) or magnetic stirrer (1000 rpm—15 min)	Indirect irradiation (10 min) with 22.8 kHz; 127 kHz; 490 kHz; 1.64 MHz or 4.6 MHz	From 70 to 400	[61]
Miglyol 812: Caprylic/capric triglycerides (10% *v/v*)	High-speed homogenization (5 min)	1.7 MHz (1 h)	220	[62]
Sunflower oil (5% *v/v*)	High-speed homogenization (5 min)	1.7 MHz (1 h)	154	[63]
Paraffin oil (8.2% *w/w*) + oleic acid (0.09% *w/w*)	High-speed homogenization (5 min)	1.7 MHz (1 h)	1920	[64]

**Table 3 foods-11-02194-t003:** Interfacial organization hypotheses defined from different experimental methods and interfaces studied.

Interface Studied	Methods	Hypotheses			
		OH^−^ Adsorption	H_3_O^+^ Adsorption	OH^−^ and H_3_O^+^ Adsorption	Other Hypotheses
Oil/Water	Surface charge	[21,22,23,24,77,81,82,83,84]	-	[25,77]	[85,86]
	Spectroscopy	[26,87]	-	[88,89]	[85,89,90,91]
	Simulation	-	-	-	[92,93]
	Surface tension	-	-	-	[94]
Air/Water	Surface charge	[77,80,95,96]	-	[77,97,98]	-
	Spectroscopy	[99,100,101]	[27,29,102,103,104,105]	[106]	[100]
	Simulation	[101,107,108,109,110,111,112]	[28,29,30,113,114,115,116,117]	[28,112,118,119,120,121]	[93,112,122]
	Surface tension	[108,109]	[30]	[120,123]	-

## Data Availability

Not applicable.

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
