# Peer review of "Review of High-Frequency Ultrasounds Emulsification Methods and Oil/Water Interfacial Organization in Absence of any Kind of Stabilizer"

_foods, 2022, doi:10.3390/foods11152194_

Round 1

Reviewer 1 Report

This manuscript firstly reviewed the numerous methods for emulsion formulations which include high energy processes such as low-frequency ultrasounds, high-pressure homogenization and microfluidization, as well as low energy processes like membrane emulsification. Then, this review focuses on the high-frequency ultrasounds process, which is thought to be able to make surfactant-free emulsion with special oil/water interfacial organization. The topic of this review is interesting, and the framework design is logical, showing some latest findings about the surfactant-free oil/water interfacial organization. However, the technical quality of the present manuscript needs to be further improved.

The title of the present manuscript needs to be revised, and “high-frequency ultrasounds emulsification methods” is suggested for consideration.

In the section of Abstract, the authors are suggested to add the basic definition for high-frequency ultrasounds, which may be clearer for readers to understand this review.  

The quality of tables 1-3 is insufficient for publication and needs to be reorganized in the revised manuscript.

Reviewer 2 Report

In this manuscript, “Review of ultrasounds emulsification methods and surfactant-free oil/water interfacial organization”, at first, the authors introduced both high and low-energy ways to fabricate emulsions and focus on the high-frequency ultrasounds method. Then the authors introduced the hypothesis of the surfactant-free interface. The paper did a good job at summarizing the stabilization, fabrication, and interface states of emulsions. However, still have many places that lack logic and connections. So, the following comments are suggested to further improve this manuscript.

Line 65: Why replace with green and natural ones is challenging? The author should give us more explanation. Because there are so many good natural emulsifiers.

Line 66-67: What is “surfactant-free emulsion formulation”? Please list the specific examples.

Introduction: As I understand, instead of surfactant, the emulsions can be stabilized by lots of molecules and particles. However, the author didn’t give a detailed introduction to them and instead directly go to their hypothesis (Line 76-81). It makes me confused. I have to say the first part of the introduction is great but the last two paragraphs lack logic and need to rewrite.

Figures: Why not use the colorful ones? And Figure 5 is difficult to understand.

What’s the connection between section 2 and section 3? And how can the conclusion from section 3 benefit the high-frequency ultrasounds? The authors should show us more and have more discussion of the connections. 

Title and the whole paper: Not sure if the author should change the “surfactant-free” to “emulsifier-free”.

Reviewer 3 Report

The manuscript reviews emulsion formation by ultrasound methods and the mechanism of interface stabilization by ions. Considering the huge literature available in the fields, the manuscript may be useful for colloid scientists working in the field. Based on this, I recommend minor revisions, as stated below.  

In the paragraph below equation 1, it should be discussed that steric repulsion, caused by polymers or macromolecules adsorbed at the interface, can also contribute to droplet stabilization, as well as the electrostatic repulsion. In the same paragraph, authors refer to coalescence as an irreversible process. Would it be the coagulation? What is the difference between coalescence and coagulation for emulsions?   I would avoid the use of "surfactant-free emulsions" for this particular case because this review deals with emulsions formed in the absence of any kind of stabilizer with interfacial activity. Pickering emulsions are also frequently named as "surfactant-free" but do not fit into the emulsions approached in the current text.  

On line 68, authors state that ultrasound generation, high-pressure homogenizers and microfluidizers are commonly used to prepare emulsions. However, most of the emulsions reported in the literature are prepared using common shearing methods (vortex, turrax, etc). This sentence should be evaluated.  

A discussion focusing on the applications of the information summarized by the review in food science should be given at the end of the text, considering the audience reached by the journal.  

English revisions are recommended in the whole manuscript.    

Reviewer 4 Report

This is an interesting review, which discusses the traditional emulsification methods and the need to switch to the new surfactant-free methods. In particular both macroscopic and microscopic evidences for the structure of the interphase and distribution of H+ and OH- ions in the hydrophobic/hydrophilic interphase are well discussed. I have two point to be be addressed:

1-Page 12, Sec. “3. Surfactant-free oil/water interface organization”, the definition of interface as “a region of space through which a system changes from one phase (liquid or solid) to another (solid, liquid, or gas), that change usually occurring over a distance of one to several tens of unit (atomic or molecular) diameter”. I think this is the definition of “interphase”. Interface is a 2D concept, while interphase is a 3D concept, therefore, the interphase has a thickness of a few molecular diameters, and the thickness depends on the property of interest. Please take care of this distinction in the whole text.

2-Sec. 3.1 and its subsections, consist of a careful review of the possibility of ion accumulation at the interface. As the authors have concluded, this subject is still a matter of debate. There exist a similar phenomenon on ion accumulation at the interface in ionic liquids and their mixtures with water, where simulation results confirm that larger size ions more accumulate at the liquid/vacuum or polar/nonpolar interfaces than the smaller size ions of the same charge. This correlates more with a higher tendency of OH- to be accumulated at the interface than H+(H3O+), which is also referred to in some places of the present review. I offer the authors to give address to this point, in case they think this might be of use to the readers of their review.

Round 2

Reviewer 2 Report

The author had addressed all my comments, so I agree to publish.